# SELF-ANCHOR: LARGE LANGUAGE MODEL REASONING VIA STEP-BY-STEP ATTENTION ALIGNMENT

## ABSTRACT

To solve complex reasoning tasks for Large Language Models (LLMs), prompting-based methods offer a lightweight alternative to fine-tuning and reinforcement learning. However, as reasoning chains extend, critical intermediate steps and the original prompt will be buried in the context, receiving insufficient attention and leading to errors. In this paper, we propose SELF-ANCHOR, a novel pipeline that leverages the inherent structure of reasoning to steer LLM attention. SELF-ANCHOR decomposes reasoning trajectories into structured plans and automatically aligns the model's attention to the most relevant inference steps, allowing the model to maintain focus throughout generation. Our experiment shows that SELF-ANCHOR outperforms SOTA prompting methods across six benchmarks. Notably, SELF-ANCHOR significantly reduces the performance gap between "non-reasoning" models and specialized reasoning models, with the potential to enable most LLMs to tackle complex reasoning tasks without retraining.

## 1 INTRODUCTION

Reasoning abilities in Large language models (LLMs) are key to solve complex tasks, from mathematical problem solving to logical inference and multi-step reasoning (Ahn et al., 2024; Huang & Chang, 2023; Cheng et al., 2025). Recent LLMs, such as OpenAI o1 (OpenAI, 2024) and DeepSeek-R1 (DeepSeek-AI, 2025), have further advanced their reasoning capabilities through fine-tuning and reinforcement learning (Luo et al., 2024; et al., 2025). Howevere, despite their impressive performance, they require substantial computation and a considerable amount of training data.

As an alternative, prompting-based methods emerged to induce LLMs' inherent reasoning capabilities at test time without updating model parameters. Methods such as Self-Refine (Madaan et al., 2023) and ReAct (Yao et al., 2023) facilitate reasoning by iteratively expanding and refining the generation process. Methods like Self-planning (Jiang et al., 2024), Plan-and-Solve (Wang et al., 2023), and Re-Reading (Xu et al., 2024) explicitly decompose complex problems before solving them. However, a tradeoff of being training-free is that these prompting-based methods necessitate long reasoning chains because they depend on iterative generation or explicit planning. While this can be seen as the cost of training-free methods, recent studies have revealed another issue: long-context reasoning can cause severe attention misalignmentt issue (Gu et al., 2024; Chi et al., 2023; Liu et al., 2024; Sun et al., 2024; Yao et al., 2021; Tian & Zhang, 2024; Li et al., 2024; Hong et al., 2025).

As a key component in LLMs, the attention mechanism (Vaswani et al., 2023) enables LLMs to selectively integrate relevant information from preceding context. However, LLM attention is an inherently limited resource. As the generation proceeds, the number of preceding tokens increases, making it increasingly difficult for the model to pay attention to the relevant context, especially when the context is long and complex (Tian & Zhang, 2024; Li et al., 2024; Hong et al., 2025). In such cases, even with the ability to correctly predict next token based on the corresponding context, LLMs may attend to irrelevant context, thereby generating off-topic or wrong results. As shown in Figure 1, the intermediate reasoning steps and most of the original prompt will be buried in the middle, receive insufficient attention, and consequently, introduce errors (Liu et al., 2024).

To mitigate such attention misalignment issues, recent works explicitly steer LLM attention to influence generation behavior. For example, PASTA (Zhang et al., 2023) adjusts the self-attention distribution within a subset of attention heads, while SPA (Tian & Zhang, 2024) simulates attention

steering through logit arithmetic. However, these methods mainly focus on developing robust attention steering mechanisms, while requiring humans to specify which tokens the model should pay more attention to. Since such tokens can vary significantly at different generation steps and across different tasks, it is unrealistic for humans to manually decide for every generation step.

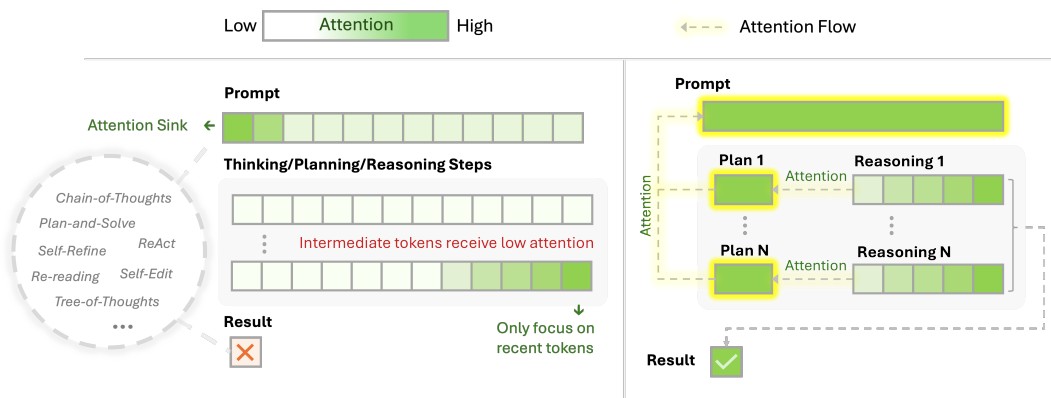

Figure 1: Comparison between existing reasoning methods and SELF-ANCHOR. Due to inherent attention patterns, existing reasoning methods may easily overlook intermediate reasoning and make mistakes. SELF-ANCHOR addresses this by decomposes the task into plans, and takes each plan as a component for attention alignment.

To reduce human efforts, we introduce SELF-ANCHOR, a novel generative pipeline that explicitly aligns LLM attention by leveraging the inherent structure of a reasoning chain. SELF-ANCHOR builds on two key insights: (1) complex reasoning problems can be decomposed into structured plans, and (2) each decomposed plan can naturally serve as a component for attention alignment. As illustrated in Figure 1, SELF-ANCHOR decomposes the original prompt into plan steps with corresponding reasoning steps. During generation, it automatically selects and steers the model attention to the prompt and the corresponding plan. This enables the LLM to keep attending to both the problem statement and immediate reasoning goals, thereby preventing attention mistakes among continuously expanding reasoning steps.

We evaluate SELF-ANCHOR on six benchmarks and six base LLMs with varying sizes, and compare it with SOTA prompting methods. The evaluation benchmarks include three mathematical reasoning benchmarks (GSM8K, AQuA, MATH), two commonsense benchmarks (StrategyQA & Things for Doing), and an multi-task evaluation benchmark (BIG-Bench Hard, BBH). SELF-ANCHOR consistently improves accuracy on all the settings, outperforming all the prompting-based baselines by at least 5.44% on average. In addition, we demonstrate that SELF-ANCHOR achieves performance on par with five reasoning models but with substantially lower cost and complexity, suggesting a practical alternative to applying reinforcement learning to enhance the reasoning capability of LLMs.

## 2 METHOD

### 2.1 SELF-ANCHOR

**Reasoning as the Scaffold for Attention Alignment.** We are inspired by the observation that planning offers a natural scaffold for attention alignment. Specifically, planning helps the model to understand and break down complex problems into subproblems (Jiang et al., 2024; Wang et al., 2023). Each decomposed plan step provides a correspondence to subsequent reasoning, which can naturally serve to guide attention alignment. Building on this idea, we design SELF-ANCHOR to leverage the inherent structure in the reasoning chain to conduct the attention alignment. SELF-ANCHOR prompts the model to decompose the prompt into plans, and then generates and corresponding reasoned solutions with explicit attention steering.

**Attention Steering Mechanism.** Several studies have explored methods for attention steering (Zhang et al., 2023; Shi et al., 2023; Tian & Zhang, 2024). These methods primarily address *how to steer the model attention*. By contrast, our work focuses on an orthogonal problem—*how to automatically select relevant context* and uses these methods as a plug-and-play component. In this work, we choose Selective Prompt Anchoring (SPA) (Tian & Zhang, 2024) as the underlying attention steering mechanism, since it is low-cost and efficient. We briefly summarize how SPA works here and refer interested readers to the original paper for more technical details.

Given a set of tokens that an LLM should pay more attention to, SPA simulates attention steering through logit arithmetic. Specifically, it estimates the influence of selected tokens by contrasting the original logits with the logits when these tokens are masked, and then add this influence back to the original logits. Formally, the steered logit is represented by the linear combination of the original logits and the logits with selected tokens masked, where $\omega_i$ is a coefficient determining the attention steering strength. Formally, it can be represented as

$$\mathbf{logits}_i^{\text{steered}} = \omega_i \cdot \mathbf{logits}_i^{\text{original}} + (1 - \omega_i) \cdot \mathbf{logits}_i^{\text{mask}} \tag{1}$$

**Selection of Attention Alignment.** Building on this attention steering mechanism, we propose a novel strategy that dynamically aligns model attention to changing context tokens during generation.

According to the attention sink phenomenon (Xiao et al., 2023), the model already attends strongly to initial tokens and the recent generated tokens. Our method complementarily steer the model attention to the planning steps when conducting the corresponding reasoning. Furthermore, SELF-ANCHOR additionally steers the model attention to the question in the prompt, which serves as the core generation purpose, ensuring the model keeps focusing on the problem statement. [1]

Formally, we use $f(x, \mathbf{S})$ to represent the generation function with attention steering, where $x$ represents the entire preceding tokens and $\mathbf{S}$ represents the selected tokens where the model's attention should be steered to. Let sys denote the system prompt together with high-level background instructions (e.g., *"You are a helpful assistant"*), and let $Q$ denote the core question under consideration. Thus, the original prompt can be represented as concat(sys, $Q$). The generation of SELF-ANCHOR consists of two parts: the *planning* and the corresponding *reasoning*.

At step $i$, the planning is generated by

$$\text{plan}_i = f\Big(\text{concat}\big(\text{sys}, Q, \text{plan}_1, \text{plan}_2, \ldots, \text{plan}_{i-1}\big), \mathbf{Q}\Big). \tag{2}$$

The planning is generated by

$$\text{reason}_i = f\Big(\text{concat}\big(\text{sys}, Q, \text{plan}_1, \text{plan}_2, \ldots, \text{plan}_{i-1}\big), \mathbf{concat}(\mathbf{Q}, \mathbf{plan_i})\Big). \tag{3}$$

The generation proceeds by alternating between $\text{plan}_i$ and $\text{reason}_i$, until the process terminates with the final result.

**Dynamic Tuning of Attention Alignment Strength.** As reasoning trajectories progress, the required degree of attention alignment may vary at different steps. According to prior work (Geng et al., 2024; Fu et al., 2025), LLMs' predicted probability distribution can be viewd as a confidence signal to determine its prediction quality. High-confidence prediction suggests reliable generation, implying a correct attention, whereas low confidence may indicate unreliable generation and attention drift. Therefore, we introduce step-level anchoring strength $\omega_i$ that are dynamically adjusted based on model confidence.[2] Let $P_i = \{p_1, p_2, \ldots, p_m\}$ represent the predicted probability at step $i$. We calculate the confidence score using the harmonic mean of $P_i$

$$p_{\text{avg}} = \frac{n}{\sum_{i=1}^{n} \frac{1}{p_i}} \tag{4}$$

This confidence score serves as additional factor to scale the attention steering strength $w_i$ in Equation 1. We discuss detailed design choices and experiments in Appendix B.

---

[1] Alternative anchoring strategies are discussed in Appendix D

[2] Our strength adjustment strategy builds upon the confidence-modulated strength strategy in SPA (Tian & Zhang, 2024). While SPA adjusts the strength based on confidence for each logit at the vocabulary level, we introduce an additional factor to adjust the strength for each step.

## 3 EXPERIMENTS

### 3.1 BENCHMARKS

We evaluated SELF-ANCHOR on six benchmarks. The first three benchmarks incorporate GSM8K (Cobbe et al., 2021), AQuA (Ling et al., 2017), and MATH (Hendrycks et al., 2021) represent arithmetic reasoning. The second two benchmarks include StrategyQA (Geva et al., 2021), and Thinking for Doing (T4D) (Zhou et al., 2023) represent commonsense reasoning. Lastly, we evaluated on a subset of BIG-Bench Hard (BBH) (Suzgun et al., 2022), which covers a diverse range of reasoning problems spanning the multi-step algorithmic reasoning, natural language understanding, the application of world knowledge, and Multilingual Knowledge. We report final answer accuracy across all benchmarks[3].

### 3.2 MODELS AND BASELINES

**Base Models.** We conduct our experiments on six non-reasoning LLMs spanning various scales. For non-reasoning models, we select Llama-3.1-8B-Instruct (Grattafiori et al., 2024), Llama-3.2-3B-Instruct (Grattafiori et al., 2024), Phi-4-mini-instruct (Abouelenin et al., 2025), Qwen3-4B-Instruct-2507 (Team, 2025), Phi-4 (Abdin et al., 2024), and Qwen3-30B-A3B-Instruct-2507 (Team, 2025).

**Comparison Baselines.** We compare our method against three representative prompting methods for LLM reasoning. First, we include **CoT** (Wei et al., 2022; Kojima et al., 2022), a widely used baseline that models are prompted to generate a reasoning process leading to the final answer. Second, we include **Plan-and-Solve+ (PS+)** (Wang et al., 2023), a prompting method that models are prompted to first generate a plan and then solve the problem. Third, we include **Re-Reading (RE2)** (Xu et al., 2024), which asks the model to read the question again and then solve the problem.

Furthermore, we consider five state-of-the-art reasoning LLMs as baselines to see if non-reasoning LLMs combined with SELF-ANCHOR achieve competitive performance against reasoning models. The reasoning models include Phi-4-mini-reasoning (Abouelenin et al., 2025), Qwen3-4B-Thinking-2507 (Team, 2025), DeepSeek-R1-Distill-Llama-8B (DeepSeek-AI, 2025), Phi-4-reasoning (Abdin et al., 2024), and Qwen3-30B-A3B-Thinking-2507 (Team, 2025).

### 3.3 MAIN RESULTS

**Mathematical Reasoning.** Arithmetic reasoning represents one of the most challenging aspects of LLM reasoning capabilities. As shown in Table 1, SELF-ANCHOR consistently improves accuracy across three arithmetic benchmarks. These gains reach over 10% improvements on GSM8K, over 5% on AQuA, and up to 8% on MATH across most models, outperforming all competing methods. While PS+ and RE2 also demonstrate potential for enhancing mathematical reasoning performance, our experiments show performance degradation on certain LLMs, suggesting limited generalization capabilities.

Interestingly, the three benchmarks span increasing difficulty levels, from grade-school problems (Cobbe et al., 2021) to GMET/GRE (Ling et al., 2017) level and competition-level problems (Hendrycks et al., 2021). SELF-ANCHOR demonstrates performance gain in all three benchmarks, suggesting superior generalization capabilities across diverse model architectures and reasoning complexity levels.

**Commonsence Reasoning.** StrategyQA requires multi-hop reasoning over commonsense knowledge. As detailed in Table 1, SELF-ANCHOR persistently improves accuracy across six evaluated LLMs. In contrast, PS+ and RE2 occasionally outperform the baseline CoT method.

For T4D, a grounded social agent reasoning task requires mental state reasoning to determine appropriate actions. SELF-ANCHOR demonstrates significant performance gains over 9% in four LLMs. In comparison, both PS+ and RE2 exhibit mixed performance; they tend to be more effective in larger models. These findings highlight the challenge of applying generic prompting strategies to specialized reasoning domains.

---

[3]Prompt templates and evaluation details are provided in Appendix E.

Table 1: Evaluation results on six benchmarks. Best results are shown in **green**, and those indicating a performance drop compared to standard greedy decoding are shown in grey.

| Model | Method | Math | | | CommonSense | | BBH |
|-------|--------|------|------|------|-------------|------|-----|
| | | GSM8K | AQuA | MATH | StrQA | T4D | |
| Llama3.2-3B | CoT | 63.84 | 46.06 | 42.5 | 66.64 | 32.98 | 35.09 |
| | PS+ | 70.62 (+6.78) | 38.58 (-7.48) | 44.50 (+2.00) | 62.79 (-3.85) | 31.81 (-1.17) | 40.51 (+5.42) |
| | RE2 | 57.38 (-6.46) | 47.28 (+1.22) | **45.00 (+2.50)** | 65.03 (-1.61) | 30.32 (-2.66) | 38.02 (+2.93) |
| | SELF-ANCHOR | **77.86 (+14.02)** | **48.43 (+2.37)** | **45.00 (+2.50)** | **67.55 (+0.91)** | **43.79 (+10.81)** | **50.48 (+15.39)** |
| Phi-4-mini-4B | CoT | 75.36 | 61.81 | 51.00 | 67.03 | 39.54 | 60.51 |
| | PS+ | 87.17 (+11.81) | 62.02 (+0.21) | 54.50 (+3.50) | 58.86 (-8.17) | 41.16 (+1.62) | 59.63 (-0.88) |
| | RE2 | 85.75 (+10.39) | 59.06 (-2.75) | 51.50 (+0.50) | 61.83 (-5.20) | 44.50 (+4.96) | 61.39 (+0.88) |
| | SELF-ANCHOR | **88.02 (+12.66)** | **68.50 (+6.69)** | **59.00 (+8.00)** | **68.69 (+1.66)** | **49.47 (+9.93)** | **62.42 (+1.91)** |
| Qwen3-4B | CoT | 86.66 | 73.62 | 82.00 | 68.03 | 70.21 | 73.33 |
| | PS+ | 82.79 (-3.87) | 71.65 (-1.97) | 83.50 (+1.50) | 68.56 (+0.53) | 59.82 (-10.39) | 70.99 (-2.34) |
| | RE2 | 79.98 (-6.68) | 75.98 (+2.36) | 82.50 (+0.50) | 69.83 (+1.80) | 67.02 (-3.19) | **75.75 (+2.42)** |
| | SELF-ANCHOR | **87.26 (+0.60)** | **79.92 (+6.30)** | **86.50 (+4.50)** | **70.13 (+2.10)** | **71.56 (+1.35)** | 75.31 (+1.98) |
| Llama3.1-8B | CoT | 61.85 | 50.79 | 44.50 | 70.24 | 26.77 | 49.45 |
| | PS+ | 62.24 (+0.39) | 48.65 (-2.14) | 47.00 (+2.50) | 65.85 (-4.39) | 28.79 (+2.02) | 51.72 (+2.27) |
| | RE2 | 57.68 (-4.17) | 51.97 (+1.18) | 44.50 (+0.00) | 69.74 (-0.50) | 28.55 (+1.78) | 54.58 (+5.13) |
| | SELF-ANCHOR | **76.72 (+14.87)** | **55.51 (+4.72)** | **52.50 (+8.00)** | **73.54 (+3.30)** | **40.01 (+13.24)** | **58.53 (+9.08)** |
| Phi-4-15B | CoT | 73.16 | 68.11 | 74.50 | 77.51 | 73.94 | 72.08 |
| | PS+ | 79.07 (+5.91) | 72.75 (+4.64) | 72.50 (-2.00) | 75.59 (-1.92) | 76.24 (+2.30) | 68.42 (-3.66) |
| | RE2 | 74.87 (+1.17) | 69.29 (+1.18) | 73.50 (-1.00) | 76.90 (-0.61) | 75.79 (+1.85) | 74.94 (+2.86) |
| | SELF-ANCHOR | **82.41 (+9.25)** | **79.13 (+11.02)** | **81.00 (+6.50)** | **77.82 (+0.31)** | **85.99 (+12.05)** | **75.31 (+3.23)** |
| Qwen3-30B | CoT | 84.46 | 81.10 | 78.00 | 78.60 | 84.92 | 67.69 |
| | PS+ | 85.11 (+0.65) | 80.83 (-0.27) | 73.50 (-4.50) | 78.70 (+0.10) | 82.62 (-2.30) | 66.67 (-1.02) |
| | RE2 | 87.21 (+2.75) | 82.28 (+1.18) | 76.00 (-2.00) | 78.91 (+0.31) | **89.36 (+4.44)** | **70.55 (+2.86)** |
| | SELF-ANCHOR | **87.41 (+2.95)** | **83.46 (+2.36)** | **87.00 (+9.00)** | **79.65 (+1.05)** | 85.56 (+0.64) | 69.30 (+1.61) |

**BBH.** BBH aggregates challenging algorithmic and symbolic tasks. SELF-ANCHOR demonstrates average performance gains ranging from 1.61% to 15.39%[4]. Among all sub-benchmarks, we find that tasks requiring tracking of intermediate reasoning benefit the most, for example *date understanding*, and *logical deduction*. We attribute this to SELF-ANCHOR's attention steering that augments critical reasoning steps and the original question throughout generation. In contrast, PS+ and RE2 show inconsistent improvements.

In summary, these results highlight two takeaways. First, while prompting strategies are effective in some tasks, they lack robustness across benchmarks and model architectures, tending to be more effective on larger LLMs and simpler reasoning tasks. This may be because larger models are more capable of following instructions and external guidance to align reasoning trajectories. Second, by integrating planning, structured reasoning, and automatic anchoring, SELF-ANCHOR achieves consistent improvements across tasks, model sizes, and architectures, demonstrating both robustness and effectiveness.

### 3.4 CAN SELF-ANCHOR RIVAL RL-ENHANCED THINKING MODEL?

Recent advances in reasoning capabilities have been dominated by reinforcement learning-enhanced "thinking" models that employ extensive internal reasoning chains during inference. However, these models are costly to fine-tune and require large-scale training data. This raises a question: *Can non-reasoning LLMs combined with* SELF-ANCHOR *achieve competitive performance against reasoning models?* To investigate this question, we compare our method applied to non-reasoning LLMs against corresponding thinking models. Since thinking models typically require longer generated contexts to support their internal reasoning, we set 1.5x larger maximum token length than the non-reasoning models in our experiments.

Table 2 presents our findings across mathematical reasoning, commonsense reasoning, and symbolic reasoning tasks. Remarkably, our method applied to non-thinking models achieves competitive or superior performance compared to RL-enhanced thinking models. Specifically, our approach closes the performance gap significantly with three arithmetic benchmarks in varying difficulties, achieving

---

[4]We detail the subtask performance in Appendix G

Table 2: Evaluation comparison with thinking models

| Model | Method | Math | | | CommonSense | | BBH |
| | | GSM8K | AQuA | MATH | StrQA | T4D | |
|---|---|---|---|---|---|---|---|
| Phi-4-mini-4B | CoT | 75.36 | 61.81 | 51.00 | 67.03 | 39.54 | 60.51 |
| | SELF-ANCHOR | 88.02 | 68.50 | 59.00 | 68.69 | 49.47 | 62.42 |
| | Reasoning | 81.27 | 60.62 | 75.00 | 66.38 | 45.04 | 59.85 |
| Qwen3-4B | CoT | 86.66 | 73.62 | 82.00 | 68.03 | 70.21 | 73.33 |
| | SELF-ANCHOR | 87.26 | 79.92 | 86.50 | 70.13 | 71.56 | 75.31 |
| | Reasoning | 83.24 | 67.32 | 87.00 | 68.31 | 73.40 | 75.34 |
| Llama3.1-8B | CoT | 61.85 | 50.79 | 44.50 | 70.24 | 26.77 | 49.45 |
| | SELF-ANCHOR | 76.72 | 55.51 | 52.50 | 73.54 | 40.01 | 58.53 |
| | Reasoning | 73.62 | 62.99 | 72.50 | 65.41 | 48.58 | 64.98 |
| Phi-4-15B | CoT | 73.16 | 68.11 | 74.50 | 77.51 | 73.94 | 72.08 |
| | SELF-ANCHOR | 82.41 | 79.13 | 81.00 | 77.82 | 85.99 | 75.31 |
| | Reasoning | 81.12 | 83.20 | 95.5 | 75.43 | 74.11 | 74.98 |
| Qwen3-30B | CoT | 84.46 | 81.10 | 78.00 | 78.60 | 84.92 | 67.69 |
| | SELF-ANCHOR | 87.41 | 83.46 | 87.00 | 79.65 | 85.56 | 69.30 |
| | Reasoning | 94.5 | 80.31 | 85.00 | 77.26 | 80.96 | 76.54 |

within 5% difference of most thinking models; On commonsense reasoning tasks and BBH, SELF-ANCHOR exceeds thinking model performance on most benchmarks and LLMs.

Noteably, we observe that thinking models demonstrate superior performance on tasks where corresponding non-reasoning models show poor baseline performance. For example, Llama3.1 and Phi-4-mini show large gaps on MATH, and Llama3.1 underperforms on AQuA and BBH. In these settings, post-training with reinforcement learning significantly boosts performance in areas where models previously performed poorly. In contrast, for tasks where non-reasoning models already demonstrate strong performance, reinforcement learning provides limited improvement. This pattern is also observed in Kirk et al. (2023). Nevertheless, SELF-ANCHOR shows consistent performance improvements across all tasks and difficulty levels.

In summary, rather than learning implicit reasoning patterns through training, our approach leverages the inherent structure in the reasoning chain for attention alignment to improve the reasoning capabilities, yielding stable improvement across varying difficulty levels without additional training cost. These findings suggest that SELF-ANCHOR can serve as an effective alternative to computationally expensive RL-enhanced reasoning.

### 3.5 WHICH TYPES OF TASKS DO SELF-ANCHOR HELP THE MOST?

To understand which types of tasks SELF-ANCHOR help the most, we analyze performance gains across task complexity. We adapt the method for quantifying task complexity from Wu et al. (2025); Jin et al. (2024), where each task is represented by an individual question in a benchmark. For each benchmark, 200 questions are randomly sampled to compute the average accuracy per task across all experimented models. Task complexity is then defined as $1 - \text{accuracy}$[5], where lower accuracy indicates higher complexity.

**Reasoning Task Complexity.** First, we examine the relationship between task complexity and SELF-ANCHOR's performance gains to understand how our method scales with task difficulty. We compare SELF-ANCHOR's performance gains against RE2, a strong baseline identified in our main results (subsection 3.3). For each task, we compute performance gain as the difference in accuracy between the two methods.

As shown in Figure 2a, the box plot summarizes the distribution of performance gains for tasks across complexity ranges. While tasks in the 0.67–0.83 complexity range show a few negative cases, the average improvement remains positive across all task complexities. Overall, SELF-ANCHOR presents performance gains spanning all complexity levels, achieving approximately 7% perfor-

---

[5]Segmenting reasoning steps and task complexity details are provided in Appendix E.

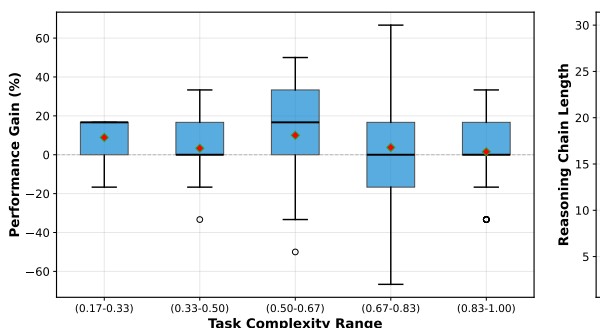 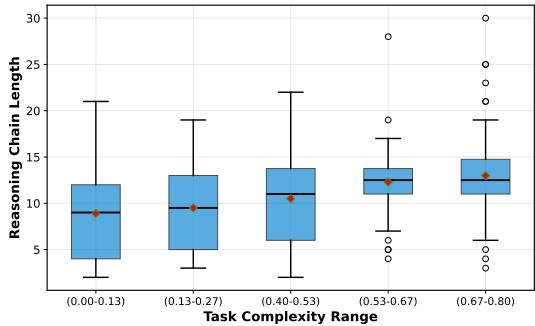

(a) Reasoning Tasks Complexity and performance gains comparison

(b) Reasoning Tasks Complexity and reasoning chain length comparison. The lower value represents the easier task.

Figure 2: Analysis of task complexity and reasoning chain length

mance improvement. These results demonstrate that SELF-ANCHOR achieves consistent gains and generalizes effectively across varying task complexities.

**Reasoning Chain Length vs. Task Complexity.** Next, we analyze how reasoning chain length scales with task difficulty. Figure 2b shows the distribution of successful reasoning chain lengths across tasks of varying difficulty on Llama3.2-3B. The results show a clear trend that as complexity increases, SELF-ANCHOR tends to generate longer reasoning chains. This aligns with the observation in Wu et al. (2025) that harder problems require longer reasoning chains to solve. We attribute this capability to the attention steering mechanism, which enables the model to focus on both problem context and immediate reasoning object throughout the reasoning, preventing attention drift as the reasoning chain extends.

In summary, our analysis highlights two takeaways: (1) SELF-ANCHOR demonstrates consistent improvements across all complexity levels, confirming its ability to generalize beyond narrow task categories. (2) SELF-ANCHOR encourage the model to generate longer reasoning chains for difficult problems, supporting its effectiveness in scaling to complex tasks.

## 3.6 EFFICIENCY

Table 3: Efficiency comparison

| Model | Token/Second | | | |
|---|---|---|---|---|
| | SELF-ANCHOR | CoT | PS+ | RE2 |
| Llama3.2-3B | 9.97 | 10.87 | 10.84 | 12.01 |
| Phi-4-mini-4B | 11.02 | 12.25 | 12.24 | 12.21 |
| Qwen3-4B | 5.41 | 6.22 | 6.22 | 6.23 |
| Llama3.1-8B | 7.54 | 8.38 | 8.26 | 8.19 |
| Phi-4-15B | 7.92 | 9.00 | 8.94 | 8.88 |
| Qwen3-30B | 1.07 | 1.39 | 1.39 | 1.37 |

SELF-ANCHOR achieved superior performance compared to state-of-the-art methods; we are committed to evaluating its efficiency. Table 3 reports inference throughput (tokens/sec) for SELF-ANCHOR compared to baseline methods. On average, SELF-ANCHOR introduces minimal computational overhead, with throughput reductions typically below 1–2 tokens/sec relative to baselines. The results highlight a favorable trade-off; SELF-ANCHOR consistently improves reasoning while maintaining comparable runtime cost. Importantly, the cost-performance balance remains stable across both smaller (3B–4B) and larger (15B–30B) models, suggesting scalability without prohibitive latency.

Table 4: Performance comparing SELF-ANCHOR with and without attention steering

| Model | Method | Math | | | CommonSense | | BBH |
|-------|--------|------|------|------|------|------|-----|
| | | GSM8K | AQuA | MATH | StrQA | T4D | |
| Llama3.1-8B | SELF-ANCHOR (w/o attention steering) | 73.01 | 53.15 | 40.90 | 71.17 | 35.28 | 52.77 |
| | SELF-ANCHOR | 76.72 | 55.51 | 52.50 | 73.54 | 40.01 | 58.53 |
| Phi-4-mini-4B | SELF-ANCHOR (w/o attention steering) | 78.77 | 58.66 | 55.50 | 68.60 | 38.83 | 57.44 |
| | SELF-ANCHOR | 88.02 | 68.50 | 59.00 | 68.69 | 49.47 | 62.42 |

Table 5: An example of BBH. Incorrect inferences are highlighted in red. Correct ones are highlighted in green.

| Prompt | *Sentence: The secretary came to the analyst's office and helped her to book a flight.*
*Options: (A) Helped the secretary book a flight (B) Helped the analyst book a flight (C) Ambiguous* |
|--------|--------|
| SELF-ANCHOR
(w/o attention steering) | { "Step 1: Identify the pronoun": "The pronoun is 'her'.",
"Step 2: Determine the antecedent of the pronoun": "The antecedent of 'her' is the subject of the sentence, which is 'The secretary'",
"Step 3: Analyze the sentence structure": The subject of the sentence is 'The secretary', and the verb is 'came'. The object of the verb 'came' is 'to the analyst's office'. The verb 'helped' has the pronoun 'her' as its object.",
"Step 4: Determine the antecedent of the pronoun based on the sentence structure": "Based on the sentence structure, the antecedent of the pronoun 'her' is the subject of the sentence, which is 'The secretary'.",
"Step 5: Choose the correct option": "The correct option is (A) Helped the secretary book a flight."} |
| SELF-ANCHOR | { "Step 1: Identify the pronoun": "The pronoun in the sentence is 'her'.",
"Step 2: Determine the antecedent of the pronoun": "To find the antecedent, we need to look for the noun that 'her' refers to. In this case, **'her' is a possessive pronoun that refers to the analyst.**",
"Step 3: Analyze the sentence structure": "The subject of the sentence is 'The secretary', and the object of the verb 'helped' is 'her'",
"Step 4: Identify the antecedent based on the sentence structure": "**'her' refers to the analyst, not the secretary.**",
"Step 5: Choose the correct option": "Since 'her' refers to the analyst, the correct option is **(B) Helped the analyst book a flight.**" } |

In summary, SELF-ANCHOR achieves strong effectiveness and efficiency, offering improved reasoning performance with only minor latency overhead compared to state-of-the-art baselines.

### 3.7 ABLATION STUDY: THE EFFECTIVENESS OF ATTENTION STEERING

To isolate the contribution of attention steering, we conducted an ablation study comparing SELF-ANCHOR with and without attention steering on two representative LLMs across all six benchmarks. As illustrated in Table 4, SELF-ANCHOR consistently outperformed its variant without attention steering, across all benchmarks, demonstrating the effectiveness of attention steering.

Table 5 further illustrates a representative example where attention steering prevents reasoning errors. In Steps 2 and 4, the SELF-ANCHOR without attention steering approach incorrectly identifies "her" as referring to "the secretary"; this may be because models over-focus on sentence subjects rather than maintaining focus on the syntactic relationships that determine pronoun reference. In contrast, SELF-ANCHOR correctly identifies that "her" refers to "the analyst" by maintaining attention on both the original question context and the current reasoning step[6].

This ablation confirms that, while structured reasoning provides a foundation for improved performance, it is often insufficient to prevent attention drift on its own. The attention anchoring component is crucial to ensure the model maintains focus throughout the reasoning process, leading to more robust and accurate results.

### 3.8 FAILURE CASE ANALYSIS

To understand the failure modes in SELF-ANCHOR. We conducted a manual failure case analysis on 200 randomly sampled cases from *casual_judgement* and *AQuA*. Our analysis identified three primary failure modes:

**Reasoning Errors (42%).** The most frequent failure mode involves LLM making mistakes during the reasoning. These include the misapplication of causal principles, flawed deductions, and incorrect conditional reasoning. For example, in one *casual_judgement* instance, the model incorrectly treated a necessary but insufficient condition as the sole causal factor, leading to an invalid conclu-

---

[6]We provide additional case studies in Appendix F for further illustration.

sion. Such errors suggest that, while attention anchoring helps maintain focus on relevant steps, it cannot fully compensate for weak logical priors or gaps in world knowledge.

**Misunderstanding the problem (36.5%).** A substantial portion of failures stems from incomprehension of complex questions, leading to errors such as misidentifying all variables, misinterpreting the problem's requirements, or incorrectly parsing the relationships between entities. For instance, in multi-variable problems, the model sometimes assigns numerical values to the wrong entity.

**Computational Errors (21.5%).** The remaining failures involve arithmetic mistakes, unit conversion errors, or algebraic slips. Even when the reasoning chain is correct, a single miscalculation often propagates to the final answer.

Taken together, these findings suggest that SELF-ANCHOR primarily mitigates *attention misalignment*, but does not fully resolve deeper issues of logical validity, semantic understanding, or computational precision.

## 4 RELATED WORK

**Prompt engineering for reasoning.** Prompt engineering has been widely adopted as a fundamental approach for enhancing LLM reasoning capabilities (Liu et al., 2023; Brown et al., 2020). A foundational line of work, initiated by Chain-of-Thought (CoT) prompting (Wei et al., 2022), encourages explicit intermediate steps, which significantly improve performance on multi-step reasoning tasks. This has inspired numerous derivatives, including problem decomposition methods (Wang et al., 2023; Zhou et al., 2022; Khot et al., 2022; Drozdov et al., 2022), as well as techniques focused on enhancing query comprehension (Xu et al., 2024; Zheng et al., 2023; Mekala et al., 2023; Deng et al., 2023; Mishra & Nouri, 2022).

While these prompting methods demonstrate effectiveness in specific domains, they rely on predetermined, static prompt formats for different tasks. On the other hand, LLMs remain sensitive to prompt variations and suffer from attention dilution during long generations (Liu et al., 2024; Li et al., 2024; Hong et al., 2025; Lu et al., 2021; Gu et al., 2024). SELF-ANCHOR addresses these limitations by integrating structured reasoning with dynamic attention steering. It goes beyond static prompting by enabling the model to recalibrate its focus on the most salient context at each step of the reasoning trajectory.

**Attention steering.** In contrast to the aforementioned prompt engineering, which devises better prompt strategies, attention steering methods directly guide LLMs during inference to emphasize the user-specified part of context. Specifically, Selective Prompt Anchoring (SPA) (Tian & Zhang, 2024) adjusts the logit probability distribution to emphasize the specified context. PASTA(Zhang et al., 2023) identifies and reweights a subset of attention heads to redirect the model's attention to user-specified parts. Selective Self-Attention (SSA) (Zhang et al., 2024) augments the softmax nonlinearity with a principled temperature scaling strategy. TOAST (Shi et al., 2023) learns feature selection modules that guide attention toward task-relevant information. However, these methods require manual specification of anchor content, limiting their adaptability to diverse reasoning contexts. Real-world applications demand automatic identification of relevant context elements across varying task requirement and reasoning patterns. SELF-ANCHOR addresses this limitation by leveraging structured intermediate representations to enable context-aware anchor selection without human intervention.

## 5 CONCLUSION

We presented SELF-ANCHOR, a lightweight pipeline that leverages the inherent structure of reasoning for attention alignment. Across six diverse reasoning benchmarks, SELF-ANCHOR consistently outperforms existing baselines. Notably, SELF-ANCHOR enhanced "non-reasoning" models achieve competitive performance with specialized reasoning models while maintaining significantly lower cost. Moreover, our analysis reveals that SELF-ANCHOR's advantages are generalizable to varying task complexities. We hope SELF-ANCHOR serves as a step toward more reliable LLMs reasoning that requires neither parameter updates nor additional sampling.

## 6 REPRODUCIBILITY STATEMENT

We have made extensive efforts to ensure the reproducibility of our work. Additional implementation details, hyperparameters, and ablation studies are provided in the appendix. We also include complete descriptions of benchmark datasets, sampling procedures, and task complexity measures.

Lastly, to foster reproducibility and further research, source code will be made publicly available upon acceptance of this paper.

## 7 ETHICS STATEMENT

This paper does not involve any ethical concerns. The proposed methods focus on improving reasoning ability and robustness in LLMs and do not raise issues related to the code of ethics.

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

## A  THE USE OF LARGE LANGUAGE MODELS (LLMS)

We leverage Large Language Models (LLMs) primarily for grammar checking and polishing for our manuscript.

## B  CALCULATING AVERAGE CONFIDENCE SCORES

We conduct an ablation study to evaluate different methods for calculating confidence scores from token-level probabilities. Let $P_i = \{p_1, p_2, \ldots, p_m\}$ represent the set of token-level confidence scores for tokens generated in the current reasoning step $i$.

We compare three approaches for calculating average confidence scores at sequence level:

**Harmonic Mean:**
$$p_{\text{harmonic}} = \frac{n}{\sum_{i=1}^{n} \frac{1}{p_i}} \tag{5}$$

**Geometric Mean:**
$$p_{\text{geometric}} = \left( \prod_{i=1}^{n} p_i \right)^{1/n} \tag{6}$$

**Arithmetic Mean:**
$$p_{\text{arithmetic}} = \frac{1}{n} \sum_{i=1}^{n} p_i \tag{7}$$

Table 6 presents the results across two reasoning benchmarks using Llama3.1 and Phi-4-mini models. The harmonic mean consistently outperforms both geometric and arithmetic means across all settings. This is because the harmonic mean is more sensitive to low confidence values, which better captures potential attention drift during reasoning. The geometric mean performs second-best, as it also penalizes low values more than the arithmetic mean, though less aggressively than the harmonic mean. The arithmetic mean shows the weakest performance, as it can be dominated by high-confidence tokens and may miss instances where attention drift occurs for specific reasoning components.

## C  SELF-ANCHOR PROMPT DETAILS

> You are an expert problem solver. Your task is to decompose the given problem into a clear, step-by-step plan, reasoning the plan and solve the problem step by step in JSON format. For each plan step, provide a key-value pair: the key is the plan step (as stated), the value is the detailed reasoning and action for that step.
> Now, implement a reasoning structure to follow step-by-step and arrive at correct answers in JSON format. Conclude with the final answer using the format: "Final answer": "<your answer>".
> Original problem: **{question}**

## D  ALTERNATIVE DESIGN

As described in Section 2.1, our primary SELF-ANCHOR design anchors attention to the original question and the current plan step during reasoning generation. We investigate an alternative design that anchors to all prior plan steps in addition to the current step and the original question. The motivation for this alternative is that maintaining attention to all previous planning steps might provide additional context for the current reasoning step.

We compare two anchoring strategies:

- **SELF-ANCHOR (Primary):** Anchors to the original question and current plan step only, where $a_i = \{Question, Plan_i\}$
- **Anchor to All:** Anchors to the original question, current plan step, and all prior plan steps, where $a_i = \{Question, Plan_1, Plan_2, \ldots, Plan_i\}$

Table 7 presents the results comparing these two approaches across AQuA and T4D benchmarks using Llama3.1-8B and Phi-4-mini models. The primary SELF-ANCHOR design consistently outperforms the alternative that anchors to all prior steps. This performance difference may be because the alternative design dilutes attention across too many anchor points, reducing focus on the most relevant current step. In contrast, the primary design maintains sharp focus on the most relevant context while avoiding attention dilution.

Table 6: Mean selection ablation study. Results show accuracy (%) on AQuA-RAT and T4D benchmarks.

| Model | Method | AQuA-RAT | T4D |
|---|---|---|---|
| LLaMA3.1-8B | CoT | 50.79 | 26.77 |
| | SELF-ANCHOR (Harmonic) | **55.51** | **40.01** |
| | SELF-ANCHOR (Geometric) | 55.11 | 35.28 |
| | SELF-ANCHOR (Arithmetic) | 54.72 | 35.99 |
| Phi-4-mini | CoT | 61.81 | 39.54 |
| | SELF-ANCHOR (Harmonic) | **68.50** | **49.47** |
| | SELF-ANCHOR (Geometric) | 67.72 | 49.11 |
| | SELF-ANCHOR (Arithmetic) | 67.71 | 48.40 |

Table 7: Alternative anchoring design comparison. Results show accuracy (%) on AQuA-RAT and T4D benchmarks.

| Model | Method | AQuA-RAT | T4D |
|---|---|---|---|
| LLaMA3.1-8B | CoT | 50.79 | 26.77 |
| | SELF-ANCHOR | **55.51** | **40.01** |
| | Anchor to All Prior Steps | 53.54 | 32.62 |
| Phi-4-mini | CoT | 61.81 | 39.54 |
| | SELF-ANCHOR | **68.50** | **49.47** |
| | Anchor to All Prior Steps | 67.32 | 47.34 |

# E    IMPLEMENTATION AND EVALUATION DETAILS

## E.1    PROMPT EXAMPLE

**Chain-of-Thought.**

> Let's think step by step.

**Plan-and-solve+.** We adopt the implementation from Plan-and-solve+(Wang et al., 2023), for mathmatical reasoning we apply prompt:

> Let's first understand the problem, extract relevant variables and their corresponding numerals, and make and devise a complete plan. Then, let's carry out the plan, calculate intermediate variables (pay attention to correct numerical calculation and commonsense), solve the problem step by step, and show the answer.

Otherwise, we use:

> Let's first prepare relevant information and make a plan. Then, let's answer the question step by step (pay attention to commonsense and logical coherence).

**Re-Reading.**

> {**Question**}
> Read the question again:
> {**Question**}

## E.2 EVALUATION DETAILS

We adopt standard metrics used in prior work (Chuang et al., 2024; Wang et al., 2023; Zhou et al., 2024), including accuracy and exact match, for AQuA, BBH, T4D, and MATH. For GSM8K and StrategyQA, we follow the factual accuracy evaluation protocol introduced by Chuang et al. (2024).

To ensure consistent answer extraction, we prompt all models to conclude their response with the phrase: "Conclude with the final answer using the format: "Final answer": "<your answer>" where <your answer> denotes either a multiple-choice option or a string answer. We then apply task-specific heuristics to extract <your answer> from the output.

For the MATH dataset, we randomly sample 200 test instances and manually verify the extracted answers to ensure correctness.

## E.3 IMPLEMENTATION DETAILS

We follow the definitions of reasoning step segmentation and task complexity from (Wu et al., 2025; Jin et al., 2024). The performance gain of a task is computed as the difference in accuracy between two methods on that task.

To measure the number of reasoning steps, we split the full reasoning chain by "\n" and remove empty lines caused by consecutive breaks ("\n\n"). The remaining lines are counted as the CoT length.

# F CASE STUDY

In this section, we provided a qualitative example to illustrate the reasoning trajectory generated by each method. The task is to determine whether a prime number can be represented by the number of days in a week. CoT 1 and RE 3 immediately arrive at an incorrect conclusion by misidentifying 7 as a non-prime number. PS+ 2 follows a more structured plan but also makes a critical error in its final conclusion, contradicting its own evidence. In contrast, SELF-ANCHOR 4 successfully decomposes the problem, correctly identifies 7 as a prime number, and reaches the correct answer by systematically executing its reasoning plan.

---

**Algorithm 1** Chain-of-Thought (CoT)

1: *Question:* Can a prime number be represented by the number of days in a week?
2: **Prompt:** Let's think step by step.
3: **Model Completion:** No, a prime number cannot be represented by the number of days in a week because the number of days in a week is 7, which is not a prime number. Prime numbers are numbers greater than 1 that have no positive divisors other than 1 and themselves. Since 7 is divisible by 1 and 7, it is not a prime number.
4: **Final answer: No**

---

# G NUMERICAL EXPERIMENT RESULT ON BIG-BENCH-HARD

---

**Algorithm 2** Plan-and-Solve (PS+)

---

1: *Question:* Can a prime number be represented by the number of days in a week?
2: **Prompt:** Let's first prepare relevant information and make a plan. Then, let's answer the question step by step (pay attention to commonsense and logical coherence).
3: **Model Completion:** Step 1: Gather Information: Prime numbers are natural numbers greater than 1 that have no positive divisors other than 1 and themselves. The number of days in a week is 7.
4: Step 2: Plan the Solution: We will first identify prime numbers less than 7 to see if any match the number of days in a week. - Then, we will determine if any prime number can represent the number of days in a week based on our findings.
5: Step 3: Execute the Plan: Prime numbers less than 7 are 2, 3, 5, and 7. - Among these, 2, 3, and 5 are less than 7 and are indeed prime numbers. - None of these prime numbers (2, 3, 5) match the number of days in a week (7).
6: Step 4: Analyze and Draw a Conclusion: Since none of the prime numbers less than 7 (2, 3, 5) match the number of days in a week (7), a prime number cannot be represented by the number of days in a week.
7: **Step 5: Final Answer: The final answer is No**

---

**Algorithm 3** Re-Reading (RE)

---

1: *Question:* Can a prime number be represented by the number of days in a week?
2: **Prompt:** Read the question again
3: **Model Completion:** No, a prime number cannot be represented by the number of days in a week because the number of days in a week is 7, which is not a prime number.
4: Prime numbers are numbers greater than 1 that have no positive divisors other than 1 and themselves.
5: The number 7 is a prime number, but the question seems to be asking if the number of days in a week (7) can be represented by a prime number, which it already is.
6: However, if the question is interpreted as asking if there is a prime number that equals the number of days in a week, the answer would still be "No" because 7 is already a prime number.
7: If the question is asking if the number of days in a week (7) can be represented by any prime number, the answer would be "Yes" because 7 itself is a prime number.
8: However, the phrasing of the question suggests that it is asking if the number of days in a week can be represented by a prime number other than 7, in which case the answer would be "No."
9: For clarity, I will assume the question is asking if the number of days in a week (7) can be represented by a prime number other than itself, and the final answer is "No."
10: **"Final answer": "No"**

---

**Algorithm 4** SELF-ANCHOR

---

1: *Question:* Can a prime number be represented by the number of days in a week?
2: **Prompt:** Decompose the given problem into step-by-step plan, reasoning the plan and solve the problem step by step in JSON format. For each plan step, provide a key-value pair: the key is the plan step, the value is the detailed reasoning for that step.
3: **Model Completion:** "Understand the question.": "The question asks if a prime number can be represented by the number of days in a week.",
4: "Identify the number of days in a week.": "There are 7 days in a week.",
5: "Determine if 7 is a prime number.": "A prime number is a natural number greater than 1 that has no positive divisors other than 1 and itself.",
6: "Check if 7 meets the criteria of a prime number.": "7 is greater than 1 and its only divisors are 1 and 7.",
7: **"Final answer": "Yes"**

---

Table 8: BBH details

| Model | Method | BBH | | | | | |
|-------|--------|-----------------|-----------------|-------------------|-------------------|---------------------------------------|--------|
| | | disambiguation_qa | causal_judgement | date_understanding | logical_deduction | salient_translation_error_detection | snarks |
| Llama3.1-8B | CoT | 100 | 91/187 | 182 | 67 | 110 | 125 |
| | PS+ | 138 | 108 | 136 | 78 | 124 | 122 |
| | re-read | 124 | 98 | 190 | 85 | 121 | 127 |
| | SELF-ANCHOR | 152 | 100 | 160 | 128 | 132 | 127 |
| Llama3.2-3B | CoT | 88/250 | 71 | 102 | 34 | 95 | 89 |
| | PS+ | 104 | 63 | 97 | 86 | 98 | 105 |
| | re-read | 74 | 84 | 127 | 40 | 89 | 105 |
| | SELF-ANCHOR | 131 | 101 | 131 | 101 | 96 | 129 |
| Phi4-mini-4B | CoT | 162/250 | 113 | 154 | 114 | 149 | 134 |
| | PS+ | 160 | 126 | 138 | 130 | 136 | 124 |
| | re-read | 162 | 115 | 165 | 113 | 135 | 148 |
| | SELF-ANCHOR | 152 | 116 | 174 | 128 | 140 | 142 |
| Qwen3-mini-4B | CoT | 183/250 | 121 | 174 | 208 | 167 | 148 |
| | PS+ | 179 | 108 | 165 | 207 | 160 | 150 |
| | re-read | 179 | 121 | 203 | 230 | 147 | 154 |
| | SELF-ANCHOR | 185 | 115 | 193 | 224 | 161 | 150 |
| Phi-4-15B | CoT | 180 | 120 | 203 | 169 | 152 | 160 |
| | PS+ | 180 | 117 | 140 | 186 | 155 | 156 |
| | re-read | 179 | 120 | 221 | 191 | 158 | 154 |
| | SELF-ANCHOR | 176 | 122 | 218 | 203 | 148 | 161 |
| Qwen3-30B | CoT | 105 | 125 | 182 | 185 | 172 | 155 |
| | PS+ | 114 | 122 | 172 | 181 | 165 | 156 |
| | re-read | 113 | 125 | 185 | 210 | 174 | 156 |
| | SELF-ANCHOR | 110 | 123 | 196 | 200 | 160 | 157 |

