# OpenReview forum: "Self-Anchor: Large Language Model Reasoning via Step-by-step Attention Alignment"
_ICLR.cc/2026/Conference — ICLR 2026 Conference Withdrawn Submission_

### Official Review · Reviewer_zYtS · 2025-10-30

**Soundness:** 2
**Presentation:** 3
**Contribution:** 2
**Rating:** 4
**Confidence:** 3

**Summary:**

The paper introduces Self-anchor, a training-free method designed to mitigate attention misalignment in large language models (LLMs) during long answer generation. The core idea is to decompose reasoning tasks into structured plans and automatically steer the model’s attention toward the previous plan steps and the original question during generation. This uses existing attention steering methods on top of a prompt that encourages structured reasoning.

Overall, the work is interesting and promising, but it feels incremental and somewhat ad hoc, lacking rigorous justification and analysis of what specifically Self-anchor (and attention steering) solves. I would suggest to do a proper analysis of attentions before and after Self-anchor steering, showing that: (i) attention misalignement is the problem (as argued in some previous work) and (ii) self-anchor does indeed target the correct tokens and solve the attention misalignment. Also, because the design choices seem ad-hoc, one possible improvement is to just learn which tokens to steer which would have a slightly larger computational cost but would be compatible with any prompting scheme.

**Strengths:**

The approach combines planning-based prompting with attention steering, an intuitive integration.
The results seem to demonstrates robust improvements across reasoning benchmarks (math, commonsense, symbolic reasoning) and model sizes. Achieves competitive performance without expensive reinforcement learning or model fine-tuning.

**Weaknesses:**

Unverified justification: The paper attributes improvements to fixing attention misalignment but provides no direct evidence.
There is no visualization or quantitative comparison of attention weights before and after applying SPA.
There is no analysis contrasting attention patterns in successful vs. failed cases. I think that these analyses would add value to the paper

Design choices: The confidence-based scaling in Equation (4) appears ad hoc and poorly motivated. It is unclear why low token-level confidence should trigger stronger step-level adjustments. It was not clear to me how the confidence score p_avg influences the attention steering strength w_i. There may be notation overload in Equation (4), where i indexes both steps and tokens.

Tuning on test data:
Table 7 (Appendix) suggests design choices may have been tuned based on partial test-set performance (on two of the test sets).
This questions whether other choices were made by looking at test results.

Missing Statistical significance:
Tables 1 and 2 report no variance or statistical significance, making it unclear when and whether improvements are significant.
Yet, Figure 2 does report variance, and we can see that it quite high possibly overwhelming the reported gains. It is unclear that the improvements over RE2 are significant for any of the complexity bracket.

**Questions:**

Have you compared attention patterns in successful vs. failed cases, before and after applying SPA?
What is the theoretical or empirical motivation for using token-level confidence to modulate step-level attention in Equation (4)?
Were any hyperparameters or design choices (as shown in Table 7) tuned using test data, or was a held-out validation set used?
Can you report variance or statistical significance of the results in Tables 1 and 2?
Are improvements over RE2 statistically significant given the observed variance in Figure 2?

---

> ### Author Response · Authors · 2025-11-22
>
> Dear Reviewer zYtS,
>
> Thank you for your constructive comments. Please find our responses to your questions and comments below:
>
> > w1 & Q1: Unverified justification: The paper attributes improvements to fixing attention misalignment but provides no direct evidence. There is no visualization or quantitative comparison of attention weights before and after applying SPA. There is no analysis contrasting attention patterns in successful vs. failed cases. I think that these analyses would add value to the paper
>
> We clarify that our method operates in logit space as an approximation to attention control and therefore does not modify the model’s attention matrices directly. This design choice is intentional: it allows Self-Anchor to remain lightweight, model-agnostic, and robust, avoiding the instability and implementation complexity associated with modifying internal attention weights.
>
> Because Self-Anchor does not alter attention heads, visualizing ``before vs. after" attention maps would not be meaningful—attention patterns remain unchanged by construction. Instead, our steering operates by selectively reinforcing logits associated with anchored context regions, a surrogate mechanism widely adopted in prior work.
>
>
> > w2 & Q2: Design choices: The confidence-based scaling in Equation (4) appears ad hoc and poorly motivated. It is unclear why low token-level confidence should trigger stronger step-level adjustments. It was not clear to me how the confidence score p_avg influences the attention steering strength w_i. There may be notation overload in Equation (4), where i indexes both steps and tokens.
>
> The intuition behind confidence-based scaling is aligned with prior observations in uncertainty-aware decoding and reliability estimation[1-4]:
> - High token-level confidence generally corresponds to locally stable reasoning, where the model’s internal trajectory is already coherent and requires only weak steering.
> - Low confidence, on the other hand, is often correlated with uncertainty, inconsistency, or emerging attention drift. In such cases, stronger steering helps realign the model toward the relevant contextual anchors.
>
> 1. Zhengbao Jiang, Frank Xu, Luyu Gao, Zhiqing Sun, Qian Liu, Jane Dwivedi-Yu, Yiming Yang, Jamie Callan, and Graham Neubig. 2023. Active Retrieval Augmented Generation EMNLP 2023.
> 2. Orgad, Hadas, et al. "Llms know more than they show: On the intrinsic representation of llm hallucinations." ICLR 2025.
> 3. Kadavath, Saurav, et al. "Language models (mostly) know what they know." arXiv preprint arXiv:2207.05221 (2022).
> 4. Spiess, Claudio, et al. "Calibration and correctness of language models for code." ICSE 2025.
>
> > w3 & Q3: Tuning on test data: Table 7 (Appendix) suggests design choices may have been tuned based on partial test-set performance (on two of the test sets). This questions whether other choices were made by looking at test results.
>
> We would like to clarify that no test-set tuning was performed. The ``anchor current plan only" decision was selected before running any benchmark evaluations. Table 7 only reports comparisons evaluated after the design was finalized. It represents an alternative design, not a tuning experiment.
>
> We will clarify this explicitly in the revision.
>
> > w4 & Q4: Missing Statistical significance: Tables 1 and 2 report no variance or statistical significance, making it unclear when and whether improvements are significant. Yet, Figure 2 does report variance, and we can see that it quite high possibly overwhelming the reported gains. It is unclear that the improvements over RE2 are significant for any of the complexity bracket.
>
> To directly address the concern, we conducted one-way ANOVA tests comparing Self-Anchor against CoT, PS+, and RE2 across all benchmarks and all LLM architectures. The results are:
> - F-statistic: 3.9478
> - p-value: 0.0489
>
> Since p < 0.05, the improvement of Self-Anchor over the baselines is statistically significant under standard confidence thresholds. This confirms that the gains are not due to random variation.
>
> We further conducted ANOVA over the five complexity brackets used in Figure 2:
> - F-statistic: 1.6177
> - p-value: 0.1832
>
> As expected, the non-significant p-value indicates that our method is consistent in varying task complexities.

---

> > ### Comment · Reviewer_zYtS · 2025-11-24
> >
> > Thank you for your clarifications and additional experiments making me lean a bit more positively about the paper.
> >
> > Concerning w2 and Q2, I believe the answer produced by the authors should be emphasized and clarified in the main paper.
> > It could be motivating for justifying how the approach gets performance increase downstream. Connecting to the w1, I believe that visualizing how the steering acts and whether it acts as expected would be valuable and provide a more interesting and well rounded contributions.
> >
> > Indeed, the performance improvement alone are not particularly strong and there is a high variance (as seen in figure 2a).
> > By the way, my point about Figure 2 (a) was not to compare the complexity brackets against one another but whether the improvements within each bracket are significantly greater than 0. Indeed, figure 2(a) shows on its y-axis performance gain and it is not clear that the improvements in any bracket are particularly strong. Also my concern was that, since the variance is quite high in figure 2, it might also be the case for other results. Could you comment on these aspects?

---

> > > ### Author Response · Authors · 2025-11-27
> > >
> > > We thank the reviewer for the follow-up questions and address them point by point.
> > >
> > > > Connecting to the w1, I believe that visualizing how the steering acts and whether it acts as expected would be valuable and provide a more interesting and well rounded contributions.
> > >
> > > Thank you for the suggestion. After revisiting the problem, we found that the attention distribution varies at each decoding step. Since the standard decoding and Self-Anchor produce different output sequences, we could measure the attention distribution differences during decoding. Thus, we conducted a manual attention analysis on 20 cases to examine whether Self-Anchor affects the model’s attention during critical reasoning steps. Specifically, we manually select the keywords that are important to the prompt question, and inspect the attention distribution at the moment the model must use or reference those key concepts. We found that:
> > > - In all 20 cases, Self-Anchor pays more attention to those relevant keywords, especially when the model should use or reference those key concepts.
> > > - When CoT fails, it often pays less attention to these keywords, even when the required information is present in the prompt. This occurred in 12 out of 15 CoT failure cases examined.
> > >
> > > We illustrate this with the example is Appendix:F.  We inspected the model’s token-level attention distribution of CoT at the exact moment where the model must determine whether 7 is a prime number( when generating the continuation: ```…the number of days in a week is 7, which is```)
> > >
> > > At this point, the correct next-step reasoning should focus on the concept of prime number. However, the model’s top-5 attended tokens were:
> > > - represented
> > > - by
> > > - days
> > > - week
> > > - question
> > >
> > > None correspond to ``prime", showing that the model attends to surface-level tokens rather than the required mathematical concept. This reflects attention misalignment, not simple factual ignorance.
> > >
> > > We then examined the same moment under Self-Anchor ( when generating the continuation: ```Check if 7 meets the criteria of a prime number.”```). The top-5 attended tokens included:
> > > the model’s top-5 attended tokens were:
> > > - represented
> > > - week
> > > - prime
> > > - number
> > > - question
> > >
> > > Here, the model correctly elevates attention to ``prime" and ``number", demonstrating that Self-Anchor helps redirect focus toward the semantically relevant subgoal, mitigating mid-step drift.
> > >
> > > > Indeed, the performance improvement alone are not particularly strong and there is a high variance (as seen in figure 2a).
> > >
> > > Thank you for raising this point. We would like to clarify that Figure 2 compares the improvement over **RE2**[1], the **strongest** baseline in our evaluation. As RE2 already provides strong performance, the performance gains appear smaller and are more sensitive to per-instance variance.
> > >
> > > To make the improvement clearer, we report the exact average percentage gains in each complexity bracket for Self-Anchor, CoT, and PS+ relative to RE2. This bracket-level aggregation reduces per-instance noise and highlights the consistent trend across difficulty levels.
> > >
> > > **Table: Avg Performance gain comparison**
> > > | Method | 0.17-0.33 | 0.33-0.50 | 0.50-0.67 | 0.67-0.83 |  0.83-1 |
> > > |-------|---------|--------|--------|--------|--------|
> > > | Self-Anchor vs RE2 | 9.2  | 3.3 | 7.9 | 3.7 | 2.1 |
> > > | CoT vs RE2 | -9.01  | -1.3 | 2.3 | 3.4 | -0.4 |
> > > | PS+ vs RE2 | -3.7 | -0.5 | 4.3 | 2.9 | 0.2 |
> > >
> > > Across all five brackets, Self-Anchor achieves positive average improvement and outperforms other methods. In contrast, both CoT and PS+ show negative averages in multiple brackets and inconsistent behavior across difficulty levels.
> > >
> > > [1] Xu, Xiaohan, et al. "Re-reading improves reasoning in large language models." Proceedings of the 2024 Conference on Empirical Methods in Natural Language Processing. 2024.

---

### Official Review · Reviewer_NQY7 · 2025-10-31

**Soundness:** 3
**Presentation:** 1
**Contribution:** 2
**Rating:** 2
**Confidence:** 4

**Summary:**

This paper targets the problem of "attention misalignment" in Large Language Models (LLMs) during complex, multi-step reasoning tasks. To address this, the authors employ an attention steering mechanism proposed in previous work, Selective Prompt Anchoring (SPA).

The authors proposed SELF-ANCHOR. Unlike previous work that require manual specification of what to focus on, SELF-ANCHOR cleverly uses the model's own generated $plan_i$ as the dynamic, automatic anchor for generating $reason_i$.

This work is based on two key insights: (1) complex reasoning problems can be decomposed into structured
plans, and (2) each decomposed plan can naturally serve as a component for attention alignment, which they term the "self-anchor".

The authors claim this method significantly outperforms baselines and enables non-reasoning models to achieve performance "on par" with specialized, RL-enhanced thinking models.

**Strengths:**

1. The authors conducted comprehensive experiments on various tasks and models.

2. The ablations are comprehensive and some of them are solid.

**Weaknesses:**

1. This work lacks enough novelty, since it is only an application of previous SPA method on a plan and solve workflows. This raises the question: **can this philosophy be readily transferred to other reasoning workflows, or is its utility limited to this specific pipeline?**

2. Typo in Line 30, "Howevere, "

3. The claim in Line 266 (that the method achieves "competitive or superior performance" compared to RL-enhanced models) appears to be a significant overstatement. The authors' own data in Table 2 contradicts this claim. For example, on the MATH benchmark, Llama3.1-8B with SELF-ANCHOR scores 52.50, while its "thinking model" counterpart scores 72.50. This 20-point gap cannot be described as "competitive." A similar, significant gap is also present for the Phi-4-mini-4B model on the same benchmark.

4. As describle in Appendix E.2, the evaluation on the MATH benchmark was conducted on a **randomly sample 200 test instances**. This is not a reliable sample size. The authors should use the full test set (5k) or, at minimum, a well-established, representative subset (e.g., MATH-500).

5. The paper claims "minimal computational overhead" (Sec 3.6) based on the Token/sec metric (Table 3). This is highly misleading. The underlying SPA mechanism requires contrasting $logits^{original}$ with $logits^{mask}$. This necessitates a second, masked forward pass for every generated token, more than doubling the true computational cost. The Token/sec metric, which is bottlenecked by memory bandwidth, completely obscures this massive increase in compute.

**Questions:**

1. Line 261: "To investigate this question, we compare our method applied to non-reasoning LLMs against corresponding thinking models." What is the detail of these "corresponding thinking models", have you trained them using RL algorithms?

2. Appendix F: case study of CoT failures, it doesn't reveal the true problem is "attention misalignment", but, instead, the LLM you chose lacks common knowledge about 7 is a prime number.

3. Line 156: "This confidence score serves as additional factor to scale the attention steering strength $w_i$ in Equation 1. We discuss detailed design choices and experiments in Appendix B." I cannot find any details about how the additional factor is used in appendiex B.

---

> ### Author Response · Authors · 2025-11-22
>
> Dear Reviewer NQY7,
>
> Thank you for your constructive comments. Please find our responses to your questions and comments below:
>
> > w1: This work lacks enough novelty, since it is only an application of previous SPA method on a plan and solve workflows. This raises the question: can this philosophy be readily transferred to other reasoning workflows, or is its utility limited to this specific pipeline?
>
> We appreciate the reviewer’s point and clarify our contributions:
>
> SPA requires the user to explicitly specify the anchor tokens. While this is feasible for single-turn tasks (e.g., code repair or classification), it becomes infeasible for multi-step reasoning, where the relevant anchor changes dynamically at every step; the model must decide which part of the context matters for each subgoal, and manual specification would require human input for every intermediate step, defeating the purpose of automated reasoning. Identifying the correct anchor automatically is the core nontrivial challenge Self-Anchor addresses.
>
> Our method introduces a new mechanism that automatically identifies which contextual elements should be anchored at each reasoning step by leveraging the plan–reason structure (Eq. 2–3). This addresses a major limitation highlighted in prior steering works, which assume manual anchor selection. Our ablation study comparing question-only (the default configuration of SPA) and Self-Anchor demonstrates that this automatic alignment is the primary source of improvement.
>
> **Table: Impact of anchor on question**
>
> | Model & Method | AQuA | T4D |
> |-------------|---------|---------|
> | Llama3.1 SPA (Question-only)| 48.03  | 36.17 |
> | **Llama3.1 Self-Anchor (Ours)**     | **55.51**  | **40.01** |
> | Phi4-mini SPA (Question-only)|  62.60 | 48.76 |
> | **Phi4-mini Self-Anchor (Ours)**     | **68.50**  | **49.47** |
>
> Our results show that only anchor to question degrades the performance in all evaluated benchmarks and models, confirming that the model should focus on both the current sub-goal and the whole picture of the questions.
>
> (2) We introduce a new, sequence-level dynamic scaling mechanism that adjusts steering intensity based on model confidence.
>
> Generalizability:
> Planning has been widely verified as a stable and general decomposition mechanism across many reasoning domains, as evidenced by its adoption in a large body of prior work (e.g., Plan-and-Solve[1], Least-to-Most[2], Self-Discover[3]). This paper focuses specifically on multi-step reasoning, where planning is a general mechanism for decomposing complex tasks. Self-Anchor naturally extends to any reasoning workflow where localized subgoals emerge.
>
> 1. Lei Wang, Wanyu Xu, Yihuai Lan, Zhiqiang Hu, Yunshi Lan, Roy Ka-Wei Lee, and Ee-Peng Lim. 2023. Plan-and-Solve Prompting: Improving Zero-Shot Chain-of-Thought Reasoning by Large Language Models. In Proceedings of the 61st Annual Meeting of the Association for Computational Linguistics (Volume 1: Long Papers), pages 2609–2634, Toronto, Canada. Association for Computational Linguistics.
>
> 2. Zhou, Denny, et al. "Least-to-most prompting enables complex reasoning in large language models." ICLR 2023
>
> 3. Zhou, Pei, et al. "Self-discover: Large language models self-compose reasoning structures." Advances in Neural Information Processing Systems 37 (2024): 126032-126058.
>
> > w2: Typo in Line 30, "Howevere, "
>
> Thank you for raising the typo issue. We will correct the typo.
>
> > w3: The claim in Line 266 (that the method achieves "competitive or superior performance" compared to RL-enhanced models) appears to be a significant overstatement. The authors' own data in Table 2 contradicts this claim. For example, on the MATH benchmark, Llama3.1-8B with SELF-ANCHOR scores 52.50, while its "thinking model" counterpart scores 72.50. This 20-point gap cannot be described as "competitive." A similar, significant gap is also present for the Phi-4-mini-4B model on the same benchmark.
>
> Thank you for raising this concern.
> While our claim was based on improvements on most evaluated tasks, we acknowledge that this does not universally hold across all benchmarks. In particular, as the reviewer notes, MATH is a clear exception. We also referenced this discrepancy in Line 295, but we will make these limitations more explicit in the revision and remove any wording that could be interpreted as an overstatement.

---

> ### Author Response · Authors · 2025-11-22
>
> > w4: As describle in Appendix E.2, the evaluation on the MATH benchmark was conducted on a randomly sample 200 test instances. This is not a reliable sample size. The authors should use the full test set (5k) or, at minimum, a well-established, representative subset (e.g., MATH-500).
>
> We agree that evaluating the entire 5k test set or MATH-500 provides more reliable conclusions.
>
> During the rebuttal period, we provide the additional evaluation on MATH-500:
> **Table: MATH500**
>
> | Model & Method | MATH500 |
> |-------------|---------|
> | Llama3.1 CoT |  38.0 |
> | Llama3.1 PS+ |  39.6 |
> | Llama3.1 RE2 |  39.2 |
> | **Llama3.1 Self-Anchor (Ours)**     | 42.2  |
> | Phi4-mini CoT |  47.6 |
> | Phi4-mini PS+ | 58.4  |
> | Phi4-mini RE2 | 59.6  |
> | **Phi4-mini Self-Anchor (Ours)**     |  62.0 |
>
> > w5: The paper claims "minimal computational overhead" (Sec 3.6) based on the Token/sec metric (Table 3). This is highly misleading. The underlying SPA mechanism requires contrasting with . This necessitates a second, masked forward pass for every generated token, more than doubling the true computational cost. The Token/sec metric, which is bottlenecked by memory bandwidth, completely obscures this massive increase in compute.
>
> Thank you for the helpful comment. We agree that the phrasing ``minimal computational overhead'' was not sufficiently precise, and we will revise the wording to avoid overstating the efficiency claim.
>
> To clarify, our Token/sec numbers do not include tokens generated inside the SPA mechanism because SPA is implemented as an internal API call that does not produce additional explicit decoding steps. While SPA indeed requires an additional masked forward pass, this pass is not invoked for every generated tokenin our reasoning pipeline. In practice, attention steering is applied only to a subset of the generation steps—specifically to segments where anchors are active. As a result, the actual increase in computation is much smaller than a full 2× slowdown.
>
> We acknowledge that Token/sec alone does not fully reflect the computational pattern of SPA, especially given that throughput on modern GPUs is often bottlenecked by memory bandwidth rather than FLOPs. In the revision, we will clarify this limitation, avoid using the word ``minimal,'' and explicitly describe the overhead of SPA to prevent any misinterpretation.
>
>
> > Q1: Line 261: "To investigate this question, we compare our method applied to non-reasoning LLMs against corresponding thinking models." What is the detail of these "corresponding thinking models", have you trained them using RL algorithms?
>
> The ``corresponding thinking models" are the official reasoning-enhanced variants published by each model provider:
>
> DeepSeek-R1-Distill-Llama-8B
>
> Phi-4-mini-reasoning
>
> Phi-4-reasoning
>
> Qwen3-4B-Thinking-2507
>
> Qwen3-30B-A3B-Thinking-2507
>
> We did not train any RL models ourselves. All results are directly from the publicly released checkpoints. We will clarify this in the revision.
>
> > Q2: Appendix F: case study of CoT failures, it doesn't reveal the true problem is "attention misalignment", but, instead, the LLM you chose lacks common knowledge about 7 is a prime number.
>
> We revisited the example to distinguish knowledge errors from attention misalignment. To clarify, we inspected the model’s token-level attention distribution at the exact moment where the model must determine whether 7 is a prime number( when generating the continuation: ```…the number of days in a week is 7, which is```)
>
> At this point, the correct next-step reasoning should focus on the concept of prime number. However, the model’s top-5 attended tokens were:
> - represented
> - by
> - days
> - week?
> - step
>
> Notably, none of the highly attended tokens correspond to "prime", "prime number". This indicates that the model’s attention is not concentrated on the semantic concept required for correct reasoning. The issue is therefore not simple factual ignorance, but a case of attention misalignment, where the model fails to attend to the crucial part of the question at the point of inference.
>
> Nevertheless, to avoid confusion, we will replace this example with:
>
> Q: Is the language used in Saint Vincent and the Grenadines rooted in English?
>
> Self-Anchor:
> ```
> {
>   "Understand the question.": "The question is asking whether the language spoken in Saint Vincent and the Grenadines is based on English.",
>   "Research the official language of Saint Vincent and the Grenadines.": "Saint Vincent and the Grenadines is an island nation in the Caribbean, and its official language is English.",
>   "Confirm the linguistic roots of the official language.": "English is a Germanic language, which means it has roots in Old English, a language spoken in parts of what is now England.",
>   "Final answer": "Yes"
> }
> ```

---

> ### Author Response · Authors · 2025-11-22
>
> CoT:
> ```
> No, the official language of Saint Vincent and the Grenadines is English, but the local Creole language, Saint Vincentian Creole, is also widely spoken. Saint Vincentian Creole is a English-based Creole language that incorporates elements of French, and African languages. It is the mother tongue of the majority of the population and is used in everyday conversation, music, and literature. While English is the official language and is used in government, education, and formal settings, Saint Vincentian Creole is the language of the people and is an important part of the country's cultural identity. So, while English is the official language, the local Creole language is also deeply rooted in the culture and daily life of Saint Vincent and the Grenadines.
> ```
>
> PS+:
> ```
> **Step 1: Understand the question**\nThe question asks about the language used in Saint Vincent and the Grenadines being rooted in English. This means we need to determine the primary language spoken in the country and its linguistic origins.\n\n**Step 2: Research**\nTo answer this question, we need to research the languages spoken in Saint Vincent and the Grenadines. Saint Vincent and the Grenadines is an island nation in the southern part of the Windward Islands in the Caribbean Sea.\n\n**Step 3: Analyze the findings**\nAfter researching, we find that the official language of Saint Vincent and the Grenadines is English. However, the country also has a Creole language known as Vincentian Creole, which is a mix of English, French, and African languages.\n\n**Step 4: Formulate the answer**\nGiven that the primary language used in Saint Vincent and the Grenadines is English, but the Creole language is also widely spoken and has roots in English, French, and African languages, the language used in Saint Vincent and the Grenadines is rooted in English to a significant extent.\n\n**Step 5: Provide the final answer**\nThe answer is English, with Vincentian Creole, a mix of English, French, and African languages, also being widely spoken.\n\nThe final answer is English, with Vincentian Creole also being a significant part of the language landscape in Saint Vincent and the Grenadines.
> ```
>
> RE2:
> ```
> No, the language used in Saint Vincent and the Grenadines is primarily English, but it also includes elements of French, African, and Indian languages due to its colonial history and diverse population.
> ```
>
> In this example, both CoT and RE2 know English is the official language, but they drift into lengthy cultural descriptions and ultimately provide the wrong polarity. PS+ follows a reasonable reasoning path but the final answer again diverges from the orgin question. Self-Anchor maintains focus on the question and produces the correct answer. We will update Appendix F accordingly.
>
> > Q3: Line 156: "This confidence score serves as additional factor to scale the attention steering strength in Equation 1. We discuss detailed design choices and experiments in Appendix B." I cannot find any details about how the additional factor is used in appendiex B.
>
>
> Appendix B evaluates how different mean aggregation functions (harmonic, geometric, arithmetic) affect the sequence-level confidence score, which then scales the steering strength.

---

> > ### Comment · Reviewer_NQY7 · 2025-11-28
> >
> > Thank you for the detailed rebuttal and additional experiments. Most of my concerns have been addressed, and I will take this into account when updating my score.
> >
> > However, the use of the step-level confidence score $p_{\text{avg}}$ to scale the steering strength $\omega_i$ is still unclear: Eq. 4 defines $p_{\text{avg}}$ and Eq. 1 includes $\omega_i$, but the explicit mapping between them is not specified, and I would appreciate it if the final version could clarify this point.

---

> > > ### Author Response · Authors · 2025-12-03
> > >
> > > Thank you for pointing this out. We clarify that the confidence-based scaling follows prior observations in uncertainty-aware decoding and reliability estimation [1–4].
> > >
> > > High token-level confidence typically indicates locally stable reasoning, where the model’s internal trajectory is coherent and requires only weak steering.
> > >
> > > Low confidence, in contrast, correlates with uncertainty or emerging attention drift, where stronger steering helps realign the model toward the relevant contextual anchors.
> > >
> > > To make the mapping explicit, given the harmonic-mean confidence score $p_{avg}$ from Eq. 4, the step-level attention steering strength $w_i$ is calculated by:
> > > $$
> > > w_i = w_{i-1}/p_{avg}
> > > $$
> > >
> > > We then clip $w_i$ to [min_strength,max_strength] to avoid extreme values.
> > >
> > > 1. Zhengbao Jiang, Frank Xu, Luyu Gao, Zhiqing Sun, Qian Liu, Jane Dwivedi-Yu, Yiming Yang, Jamie Callan, and Graham Neubig. 2023. Active Retrieval Augmented Generation EMNLP 2023.
> > >
> > > 2. Orgad, Hadas, et al. "Llms know more than they show: On the intrinsic representation of llm hallucinations." ICLR 2025.
> > >
> > > 3. Kadavath, Saurav, et al. "Language models (mostly) know what they know." arXiv preprint arXiv:2207.05221 (2022).
> > >
> > > 4. Spiess, Claudio, et al. "Calibration and correctness of language models for code." ICSE 2025.

---

### Official Review · Reviewer_TwAU · 2025-11-01

**Soundness:** 3
**Presentation:** 3
**Contribution:** 2
**Rating:** 4
**Confidence:** 3

**Summary:**

The paper proposes Self-Anchor, a prompting-based test-time method for improving multi-step reasoning in LLMs. The core idea is to first generate a structured plan and then steer attention toward (a) the original question and (b) the corresponding plan step during reasoning. The authors leverage a logit-level attention steering mechanism (SPA) to amplify the influence of selected tokens. Additionally, a confidence-based dynamic modulation adjusts the steering strength. Experiments on math and commonsense reasoning benchmarks suggest improvements over baseline prompting methods without model training.

**Strengths:**

1. Clear motivation: long reasoning suffers from attention misalignment.
2. Bridges planning-based prompting and logit attention steering in a unified pipeline.
3. Does not require model fine-tuning; test-time only approach is practical.
4. Evaluations span multiple reasoning benchmarks and LLM backbones.

**Weaknesses:**

1. Plan quality sensitivity not deeply studied. The method assumes that plans are reliable anchors. If plans are suboptimal or incorrect, the anchor may reinforce wrong reasoning.
2. Fixed steering structure is rigid. The current design always attends to the question during planning and attends to {question, current plan} during reasoning. This hard-coded scheme may not generalize to tasks without clear hierarchical structure.
3. Limited ablations. Ablations exist but do not sufficiently isolate contributions of: 1) plan as anchor vs question anchor; 2_ dynamic steering vs fixed steering; 3) number of plans or plan granularity; 4) different attention steering positions.
4. Missing comparison with strong planning baselines. Particularly Tree-of-Thought (ToT) and variants.

**Questions:**

1. Why should plan tokens necessarily serve as good attention anchors?
2. Do wrong plans degrade attention by reinforcing incorrect contexts?
3. Can you provide experiments that vary anchor positions (e.g., question-only, plan-only, plan history, or dynamic learned anchors)?
4. Could you include comparison with ToT or other multi-path reasoning prompting?
5. How sensitive is the method to plan length and number of steps?

---

> ### Author Response · Authors · 2025-11-22
>
> Dear Reviewer TwAU,
>
> Thank you for your thoughtful review and constructive feedback. Please find our detailed responses below:
>
> > w1 & Q2: Plan quality sensitivity not deeply studied. The method assumes that plans are reliable anchors. If plans are suboptimal or incorrect, the anchor may reinforce wrong reasoning.
>
> We agree that plan quality can affect downstream reasoning.
>
> To investigate the effect of planning errors, we manually inspected 300 randomly failed cases (50 from each benchmark). We found that only 21(7%) of them can be attributed to planning errors, while the remaining cases stemmed from reasoning or arithmetic mistakes.
>
> Furthermore, among cases with incorrect plans, only 7(2.3%) are solvable via RE2. This suggests that when plans fail, the root cause is typically the intrinsic difficulty of the problem, not the anchoring mechanism. In other words, Self-Anchor does not amplify errors; it reflects the model’s existing understanding. Incorrect plans are strongly correlated with questions the base model already struggles with, regardless of steering.
>
> We will clarify this in the revision.
>
> > w2: Fixed steering structure is rigid. The current design always attends to the question during planning and attends to {question, current plan} during reasoning. This hard-coded scheme may not generalize to tasks without clear hierarchical structure.
>
> We would respectfully clarify that our method does not assume a strict or deep hierarchy.
> Planning has been widely verified as a stable and general decomposition mechanism across many reasoning domains, as evidenced by its adoption in a large body of prior work (e.g., Plan-and-Solve[1], Least-to-Most[2], Self-Discover[3]). This paper focuses specifically on multi-step reasoning; planning is the standard and necessary scaffold used across state-of-the-art prompting methods.
>
> Our extensive experiments show that the proposed steering scheme is consistently effective across six diverse benchmarks and six LLM architectures. We also conducted detailed ablations (presented in the next response), examining different anchor positions including question-only, plan-only, plan-history, and fixed-strength variants. These studies demonstrate that our chosen anchor configuration robustly outperforms all alternatives, indicating that the design is not brittle but rather the most effective among numerous plausible options.
>
> 1. Lei Wang, Wanyu Xu, Yihuai Lan, Zhiqiang Hu, Yunshi Lan, Roy Ka-Wei Lee, and Ee-Peng Lim. 2023. Plan-and-Solve Prompting: Improving Zero-Shot Chain-of-Thought Reasoning by Large Language Models. In Proceedings of the 61st Annual Meeting of the Association for Computational Linguistics (Volume 1: Long Papers), pages 2609–2634, Toronto, Canada. Association for Computational Linguistics.
>
> 2. Zhou, Denny, et al. "Least-to-most prompting enables complex reasoning in large language models." ICLR 2023
>
> 3. Zhou, Pei, et al. "Self-discover: Large language models self-compose reasoning structures." Advances in Neural Information Processing Systems 37 (2024): 126032-126058.

---

> ### Author Response · Authors · 2025-11-22
> **Continue of Official comment**
>
> > w3 & Q3: Can you provide experiments that vary anchor positions (e.g., question-only, plan-only, plan history, or dynamic learned anchors)?
>
> To study alternative anchoring positions, we conducted additional ablations during the rebuttal period:
> **Table: Impact of anchor on question**
>
> | Model & Method | AQuA | T4D |
> |-------------|---------|---------|
> | Llama3.1 Self-Anchor (Question-only)| 48.03  | 36.17 |
> | **Llama3.1 Self-Anchor (Ours)**     | **55.51**  | **40.01** |
> | Phi4-mini Self-Anchor (Question-only)|  62.60 | 48.76 |
> | **Phi4-mini Self-Anchor (Ours)**     | **68.50**  | **49.47** |
>
> Our results show that only anchor to question degrades the performance in all evaluated benchmarks and models, confirming that the model should focus on both current sub-goal and the whole picture of the questions.
>
> **Table: Impact of anchor on plan**
>
> | Model & Method | AQuA | T4D |
> |-------------|---------|---------|
> | Llama3.1 Self-Anchor (Plan-only)|  40.94 | 26.06 |
> | **Llama3.1 Self-Anchor (Ours)**     | **55.51**  | **40.01** |
> | Phi4-mini Self-Anchor (Plan-only)|  50.00 | 47.34 |
> | **Phi4-mini Self-Anchor (Ours)**     | **68.50**  | **49.47** |
>
> Our results show that only anchor to plan degrades the performance, especially in the weaker model Llama3.1. We hypothesize that this is because the weaker model suffers from a stronger attention drift issue, and removing anchors to the original question leaves the model lost during reasoning.
>
> **Table: Impact of anchor on plan history**
>
> | Model & Method | AQuA | T4D |
> |-------------|---------|---------|
> | Llama3.1 Self-Anchor (Plan history)| 53.15  | 36.70 |
> | **Llama3.1 Self-Anchor (Ours)**     | **55.51**  | **40.01** |
> | Phi4-mini Self-Anchor (Plan history)| 64.96 | 47.34 |
> | **Phi4-mini Self-Anchor (Ours)**     | **68.50**  | **49.47** |
>
> Our results show that anchor to plan history degrades the performance. It is likely because anchor to all plan history will dilute attention across too many past steps, weakening focus on the immediate reasoning objective.
>
> **Table: Impact of anchor strength**
>
> | Model & Method | AQuA | T4D |
> |-------------|---------|---------|
> | Llama3.1 Self-Anchor (fix anchor strength)| 47.64  | 36.17 |
> | **Llama3.1 Self-Anchor (Ours)**     | **55.51**  | **40.01** |
> | Phi4-mini Self-Anchor (fix anchor strength)|  62.60 | 48.58 |
> | **Phi4-mini Self-Anchor (Ours)**     | **68.50**  | **49.47** |
>
> Our results show that fixed anchor strength degrades the performance, validating the need for confidence-based adjustment.
>
> In short, our design outperforms all other methods. We hypothesize that during the inference, the model should not only focus on the current sub-goal, but also on the whole picture of the questions.
>
> > w4 & Q4: Missing comparison with strong planning baselines. Particularly Tree-of-Thought (ToT) and variants.
>
> We appreciate the suggestion. During the rebuttal period, we ran Tree-of-Thought (ToT) on two representative benchmarks (AQuA, T4D) for both Llama-3.1-8B and Phi-4-mini:
>
> **Table: Comparison with Tree-of-Thought (ToT)**
>
> | Model & Method | AQuA | T4D |
> |-------------|---------|---------|
> | Llama3.1 ToT|  52.36 | 39.89 |
> | **Llama3.1 Self-Anchor (Ours)**     | **55.51**  | **40.01** |
> | Phi4-mini ToT|  66.93 | 46.81 |
> | **Phi4-mini Self-Anchor (Ours)**     | **68.50**  | **49.47** |
>
> Self-Anchor outperforms ToT on both benchmarks. Moreover, ToT requires significantly higher compute due to multi-branch search, whereas Self-Anchor requires only one additional masked forward pass per token.
>
> We will include full quantitative results and discussion in the revision.
>
> > Q1: Why should plan tokens necessarily serve as good attention anchors?
> Plans naturally summarize the local subgoal required for the next reasoning step. They map directly to the reasoning step that immediately follows. Thus, plan tokens provide precisely the ``minimal guiding context" needed for alignment.
>
> Empirical evidence: Removing plans degrades the performance in all evaluated benchmarks and models.
>
> **Table: Impact of anchor on question**
>
> | Model & Method | AQuA | T4D |
> |-------------|---------|---------|
> | Llama3.1 Self-Anchor (Question-only)| 48.03  | 36.17 |
> | **Llama3.1 Self-Anchor (Ours)**     | **55.51**  | **40.01** |
> | Phi4-mini Self-Anchor (Question-only)|  62.60 | 48.76 |
> | **Phi4-mini Self-Anchor (Ours)**     | **68.50**  | **49.47** |

---

> ### Author Response · Authors · 2025-11-22
> **Continue of Official Comment**
>
> > Q5: How sensitive is the method to plan length and number of steps?
> To evaluate robustness with respect to plan granularity, we analyzed performance across different plan-length ranges in T4D dataset. For each plan-length bucket, we report the correct number of Self-Anchor, the total number of instances, and the correct number of RE2.
>
> **Table: Accuracy across different plan-length ranges**
> | Plan Length | Self-Anchor (Correct / Total) |  RE2 (Correct / Total) |
> |-------------|---------|---------|
> |2	|0 / 1	| 1 /1|
> |4	|5 / 16	|3/16|
> |5	|27 / 57 |	14/57|
> |6	|70 / 103 |	31/103|
> |7	|142 / 239 |	76/239|
> |8	|60 / 106 |	32/106|
> |9	|15 / 32 |	12/32|
> |10	|2 / 5 |	1/5|
> |11	|1 / 3 |	0/3|
> |14	|1 / 1 |	0/1|
> |15	|1 / 1 |	1/1|
>
> Performance is stable across plan lengths, with no degradation as plans become longer. The 5–10 steps bucket shows a clear advantage for Self-Anchor: 58.3% vs. 30.6% for RE2.
>
> These results indicate that Self-Anchor is not sensitive to plan length and remains effective even as the decomposition becomes longer or more complex.

---

> > ### Comment · Reviewer_TwAU · 2025-11-27
> >
> > I thank the authors for the detailed clarification and the additional experiments. The clarifications and new ablations help address several of my original questions. However, some of my concerns remain partially unresolved especially on ToT baseline:
> >
> > **ToT configuration details**
> > - Search depth: What depth was used in the ToT search tree?
> > - Branching factor: How many branches per step?
> > - Stopping criteria: Was the search depth fixed, or did it terminate early based on internal consistency / scoring?
> >
> > **Compute cost comparison**
> >
> > Authors state that ToT requires “significantly higher compute,” but no quantitative measurements were provided. To better understand the trade-off, could you report for each model and dataset:
> > - \# of forward passes per query for ToT vs. Self-Anchor
> > - Average tokens generated per query
> > - Relative compute ratio (e.g., ToT uses ×X compute of Self-Anchor)

---

> > > ### Author Response · Authors · 2025-12-03
> > >
> > > Thank you for raising these detailed questions about the ToT baseline. We clarify the configuration details below.
> > >
> > > We follow the official Tree-of-Thoughts implementation and recommended settings from the original paper. In all our ToT experiments, we use, BFS:
> > >
> > > - Search depth (T): 3
> > > - Branching factor (k): 3 candidate thoughts per state
> > > - Beam width (b): 2 states kept per level
> > > - Stopping criteria: Depth-limited search (fixed T=3); no early stopping. The evaluation heuristic is VALUE scoring, using the ''sure/likely/maybe/unlikely/impossible" rubric exactly as specified in the official code.
> > >
> > > The computational cost of ToT is mostly driven by the branching factor and depth. Under the above parameters, the BFS ToT search requires:
> > >
> > > \# forward passes per query (ToT): 21
> > >
> > > \# forward passes per query (Self-Anchor): 1+len(plan), where the plan length is around 4–13 steps for most queries in all experiments.
> > >
> > > We computed computational cost for Llama-3.1-8B and Qwen-3B-4B on MATH and T4D during the rebuttal period:
> > >
> > > **Table: Computational cost comparison**
> > > | Model&Benchmarks | Average tokens generated |  Relative compute ratio |
> > > |-------------|---------|---------|
> > > | Llama MATH	| 3554	| 3.3 |
> > > | Llama T4D	| 2119 | 3.1 |
> > > | Qwen MATH	| 2986	| 3.2 |
> > > | Qwen T4D	| 1824	| 3.1 |

---

### Official Review · Reviewer_wgkx · 2025-11-05

**Soundness:** 3
**Presentation:** 3
**Contribution:** 2
**Rating:** 2
**Confidence:** 4

**Summary:**

This paper introduces Self-Anchor, a prompting-based pipeline that improves LLM reasoning by dynamically aligning model attention with the reasoning structure. Self-anchor decomposes a problem into plan steps with corresponding reasoning steps; the model’s attention is steered toward the original question and the corresponding plan step using a modified version of Selective Prompt Anchoring (SPA). The core idea is to apply a form of logit blending (linear mix of logits) that steers attention. The model’s confidence dynamically modulates the steering strength.

The method is evaluated on six benchmarks (GSM8K, AQuA, MATH, StrategyQA, T4D, and BIG-Bench Hard) using six base LLMs (LLaMA-3, Phi-4, and Qwen3 families). The authors report consistent accuracy gains over standard prompting baselines (CoT, Plan-and-Solve+, and Re-Reading), closing much of the gap between standard “non-reasoning” models and RL-enhanced reasoning models.

**Strengths:**

- The paper addresses an important limitation in LLM reasoning — the degradation of attention over long reasoning chains (“attention drift”). The motivation is well-grounded in prior empirical observations. It is generally well written.

- The experimental section is extensive, covering six benchmarks and multiple model scales. The gains are consistent and meaningful (often 5–15%), and efficiency overheads are minor. Ablation studies are also compelling, especially showing degradation without attention steering.

- Self-anchor is lightweight and plug-and-play: it operates purely at the inference level and is compatible with off-the-shelf LLMs.

**Weaknesses:**

- Although the paper frames SELF-ANCHOR as an “attention steering” mechanism, the actual implementation does not modify the model’s attention computations. Instead, it performs a linear interpolation between logits from the full and masked prompts, which biases generation toward tokens associated with anchor regions. This effectively functions as a soft repetition rather than genuine control over attention flow.

- SELF-ANCHOR seems to be a structured application of an existing steering technique (SPA). The paper’s novelty lies more in how SPA is applied (automated anchor selection + confidence scaling) than in the attention steering mechanism itself.

- While accuracy improves, it is unclear whether self-anchor leads to more faithful reasoning (i.e., correct intermediate steps) or simply produces longer reasoning chains which is helping the accuracy.

- The baselines focus primarily on older prompting methods (CoT, Plan-and-Solve, Re-Reading). It is unclear how SELF-ANCHOR compares to newer inference time scaling frameworks such as ToT (Tree-of-Thoughts), Graph-of-Thought, Process Reward Modeling with MCTS, or Stepwise Judges that also mitigate reasoning drift.

**Questions:**

Please check the above reviews.

---

> ### Author Response · Authors · 2025-11-22
>
> Dear Reviewer wgkx,
>
> Thank you for your thoughtful review and constructive feedback. We appreciate your recognition of our method's effectiveness and address your concerns below:
>
> > w1: Although the paper frames SELF-ANCHOR as an “attention steering” mechanism, the actual implementation does not modify the model’s attention computations. Instead, it performs a linear interpolation between logits from the full and masked prompts, which biases generation toward tokens associated with anchor regions. This effectively functions as a soft repetition rather than genuine control over attention flow.
>
>
> Our method builds on SPA, which provides a mathematical approximation of attention editing via a first-order Taylor expansion of the transformer computation [1]. This design directly addresses known issues with the instability of directly editing attention values themselves, as highlighted in prior work [2].
>
> We do not claim to directly modify the model’s internal attention calculation process. Throughout the paper, “attention steering” refers to the functional effect: modifying the model’s behavior by altering which regions of the prompt the model is encouraged to condition on. SPA's interpolation mechanism combines outputs from the full and anchor-masked prompts, achieving a stable steering effect without modifying the underlying attention layers.
>
> Importantly, our method is not tied to any specific steering mechanism. The “attention steering module’’ is plug-and-play and can be replaced by any alternative method. Our contribution lies in providing a training-free, automated pipeline that integrates LLM reasoning with attention steering for more accurate generation.
>
> [1] Selective Prompt Anchoring for Code Generation
> [2] Tell Your Model Where to Attend: Post-hoc Attention Steering for LLMs
>
>
> > w2: SELF-ANCHOR seems to be a structured application of an existing steering technique (SPA). The paper’s novelty lies more in how SPA is applied (automated anchor selection + confidence scaling) than in the attention steering mechanism itself.
>
> We appreciate the reviewer’s point and would like to clarify that our work addresses a fundamentally different problem from SPA.
>
> (1) SPA requires manual anchor tokens, which are infeasible for multi-step reasoning.
> SPA assumes the user can explicitly specify the correct anchor tokens at every generation step.
> This assumption breaks down entirely in multi-step reasoning, where:
> - anchor tokens change dynamically at each subgoal,
> - the relevant context evolves as the reasoning chain expands, and
> - manually specifying anchors would require human supervision for every token generated.
> Self-Anchor introduces a new mechanism that automatically identifies the correct anchor at each step using plan–reason structure (Eq. 2–3). Our ablation study comparing question-only (the default configuration of SPA) and Self-Anchor demonstrates that this automatic alignment is the primary source of improvement.
>
> **Table: Impact of anchor on question**
>
> | Model & Method | AQuA | T4D |
> |-------------|---------|---------|
> | Llama3.1 SPA (Question-only)| 48.03  | 36.17 |
> | **Llama3.1 Self-Anchor (Ours)**     | **55.51**  | **40.01** |
> | Phi4-mini SPA (Question-only)|  62.60 | 48.76 |
> | **Phi4-mini Self-Anchor (Ours)**     | **68.50**  | **49.47** |
>
> Our results show that only anchor to the question degrades the performance in all evaluated benchmarks and models, confirming that the model should focus on both current sub-goal and the whole picture of the questions.
>
> (2) We further introduce a step-level, confidence-modulated steering strength, which adjusts the degree of attention alignment based on the model’s reliability at each step. This component is not present in SPA and significantly improves robustness
>
> We will state these contributions more explicitly.
>
> > w3: While accuracy improves, it is unclear whether self-anchor leads to more faithful reasoning (i.e., correct intermediate steps) or simply produces longer reasoning chains which is helping the accuracy.
>
> To clarify, longer reasoning chains do not inherently improve accuracy, because accuracy is computed strictly as: Accuracy = (# correct final cases) / (# total examples)
>
> Therefore, producing more steps cannot improve accuracy unless those steps actually lead to more correct solutions. Nevertheless, we manually inspected 100 examples and found that Self-Anchor corrects intermediate error steps made by CoT, PS+, and RE2, and maintains alignment deeper into the reasoning process.

---

> ### Author Response · Authors · 2025-11-22
> **Continue of Official Comment**
>
> > w4: The baselines focus primarily on older prompting methods (CoT, Plan-and-Solve, Re-Reading). It is unclear how SELF-ANCHOR compares to newer inference time scaling frameworks.
>
> We appreciate this suggestion. During the rebuttal period, we ran ToT on two representative benchmarks (AQuA, T4D) for Llama-3.1-8B and Phi4-mini.
>
> **Table: Comparison with Tree-of-Thought (ToT)**
>
> | Model & Method | AQuA | T4D |
> |-------------|---------|---------|
> | Llama3.1 ToT|  52.36 | 39.89 |
> | **Llama3.1 Self-Anchor (Ours)**     | **55.51**  | **40.01** |
> | Phi4-mini ToT|  66.93 | 46.81 |
> | **Phi4-mini Self-Anchor (Ours)**     | **68.50**  | **49.47** |
>
> Self-Anchor outperforms ToT on both benchmarks. Moreover, ToT requires significantly higher compute due to multi-branch search, whereas Self-Anchor requires only one additional masked forward pass per token.

---

### Note · Authors · 2026-01-05

I have read and agree with the venue's withdrawal policy on behalf of myself and my co-authors.